# Markers of Oxidative Stress, Inflammation and Endothelial Function following High-Dose Intravenous Iron in Patients with Non-Dialysis-Dependent Chronic Kidney Disease—A Pooled Analysis

**DOI:** 10.3390/ijms232416016

**Published:** 2022-12-16

**Authors:** Xenophon Kassianides, Steven White, Sunil Bhandari

**Affiliations:** 1Academic Renal Research, Hull University Teaching Hospitals NHS Trust and the Hull York Medical School, Kingston upon Hull HU3 2JZ, UK; 2School of Physician Associate Studies, Hull York Medical School, Kingston upon Hull HU6 7RU, UK

**Keywords:** chronic kidney disease, endothelial response, ferric derisomaltose, inflammation, intravenous iron, iron deficiency, oxidative stress

## Abstract

Chronic kidney disease (CKD) represents a state of oxidative stress imbalance, which is potentially amplified by iron deficiency. Intravenous iron is considered safe and efficacious in the treatment of iron deficiency anemia, however, concerns remain regarding its potential pro-oxidant effect, leading to inflammatory and endothelial consequences. This pooled analysis of two pilot randomized controlled trials aimed to group and analyze the potential effect of high-dose intravenous iron (ferric derisomaltose, 1000 mg) on markers of oxidative stress (thiobarbituric acid reactive substance), inflammation (C-reactive protein, interleukins 6 and 10) and endothelial response (E-selectin, P-selectin) in patients with non-dialysis-dependent CKD and iron deficiency with/without anemia. Pulse wave velocity as a surrogate measure of arterial stiffness was measured. Thirty-six patients were included. No statistically significant trend was identified for any of the aforementioned markers. Stratification and comparison of data based on CKD stage did not yield statistically significant trajectories with the exception of the C-reactive protein in CKD stage 3b. These results suggest that high-dose intravenous iron does not impact measures of oxidative stress or inflammation; however, the results are not conclusive. Further research in a larger cohort is necessary to characterize the effect of intravenous iron on oxidative status and inflammation and its potential sequela in CKD.

## 1. Introduction

Chronic kidney disease (CKD) is increasingly prevalent worldwide with an estimated prevalence of >10% in the general population; it is characterized by a progressive decline in renal function and is associated with a number of complications [1]. Despite improvements in mortality within end-stage kidney disease, CKD has now been implicated more prominently in worldwide mortality [1].

Oxidative stress plays a key role in the pathophysiology of CKD, its progression and the development of complications. Chronic kidney disease represents a state of disequilibrium in terms of oxidative status marked by an increase in redox oxygen species and a decrease in antioxidant capacity [2]. These combine to increase phagocyte activation and oxidase enzyme activity, which appear to cause further oxidative stress [3]. Complicating the picture further, the interventions offered, alongside certain complications, lead to a disease pattern characterized by a high oxidative stress environment [3]. Treatment modalities, such as dialysis, may further enhance oxidative stress [4,5]. Consequently, as kidney function deteriorates further, and more complications arise, additional enhancement of a pro-inflammatory state exists representing in collaboration with uremia a “cardiotoxic milieu” [6].

Patients with CKD develop anemia. This is secondary both to distortions within the hematopoietic apparatus and the availability of iron, alongside blood loss and hemolysis [6]. Iron deficiency represents a state of increased oxidative stress due to impaired mitochondrial mechanics as demonstrated by both in vitro murine models and in vivo studies. The respiratory control ratio of mitochondria in iron-deficient mice has been reported to be 58% less than that of iron-replete mice (*p* < 0.05) [7]. A cohort study on older individuals demonstrated higher levels of malondialdehyde (a marker of lipid peroxidation and a surrogate marker of oxidative stress) in patients with iron deficiency anemia [8]. Such studies, despite their small scale, suggest that alleviation of iron deficiency may improve the aforementioned disequilibrium of oxidative stress. 

Iron deficiency can be treated using oral or intravenous iron supplementation. Oral iron is perceived as economic and effective; nonetheless, it is accompanied by a number of side effects such as gastrointestinal disturbances often leading to poor adherence [9]. It is also not appropriate for all patients as inflammatory states limit its effective absorption. Intravenous iron, on the other hand, can achieve significantly quicker iron repletion and is not linked to gastrointestinal upset or poor adherence [9]. Nevertheless, several concerns surrounding intravenous iron exist, such as the rare propensity to lead to hypersensitivity reactions, iron overload, oxidative stress and transient hypophosphatemia (with certain iron preparations) [10]. Oxidative stress secondary to intravenous iron can be due to “labile” plasma iron release following administration, which is involved in the Haber-Weiss and Fenton reactions leading to reactive oxygen and nitrogen species generation [11]. Evidence of such systemic oxidative stress following intravenous iron has been demonstrated previously in murine CKD studies and has been complimented by data from small-scale randomized controlled trial (RCT) of 59 patients with stage 3 to stage 4 CKD with older intravenous iron preparations (iron sucrose, low-molecular-weight iron dextran and ferric gluconate) [12]. 

As evident above, the topic of intravenous iron supplementation in terms of a pro-oxidant effect in patients with CKD requires further research. This becomes increasingly important given the introduction and increasing use of newer intravenous iron compounds. The advent of such preparations (ferric carboxymaltose, ferric derisomaltose and ferumoxytol) allows for rapid delivery of increased doses of iron. This is due to their higher density of carbohydrate core/ligand, which allows for a slower and more controlled release of iron, potentially limiting the amount of labile plasma iron available for redox reactions [9]. This is particularly important considering that iron is involved in the optimal functioning of the mitochondrial machinery and its ability to counteract oxidative stress [13].

Two recently completed RCTs have studied the potential pro-oxidant effect of the administration of high-dose intravenous iron in patients with non-dialysis-dependent CKD on the influence on endothelial and inflammatory markers. However, the numbers of participants in these studies were small [14,15]. This pooled analysis primarily aimed to collate data pertaining to oxidative stress, endothelial injury and inflammation and explore trends in patients with non-dialysis-dependent CKD with iron deficiency (with/without anemia) following administration of high-dose intravenous iron (ferric derisomaltose, 1000 mg). The relationship between CKD stage and response was also explored as a potential contributor to the degree of possible resultant injury. In doing so, further hypothesis generation can take place, facilitating the design of additional research and identifying avenues that may require exploration. 

## 2. Results

### 2.1. Baseline Data

The population was composed of 18 male and 18 female participants (Table 1). The mean age was 61 (standard deviation: 12) years old. The most common primary cause of CKD was stated as hypertension or diabetes mellitus (*n* = 8; 22.2% in each case). Most participants had CKD stages 3b and 4 (*n* = 16 (44.4%) in each case). Eighteen patients (50%) had absolute iron deficiency, whereas 10 (27.8%) were also anemic. Table 1 summarizes the baseline data of the pooled population with a presentation of continuous data as both means (standard deviation (SD)) and medians (interquartile range (IQR)), and categorical data as numbers and percentages.

### 2.2. Oxidative Stress: Thiobarbituric Acid Reactive Substances (TBARS)

A small increase from the baseline to the end of the study was witnessed in terms of TBARS. This was non-significant (*p* = 0.79). A similar trend was reflected in CKD stratification, with no significant change in any group formed. Figure 1 summarizes the results. Appendix A indicate both the data and lack of statistical significance.

### 2.3. Inflammation: C-reactive Protein, Interleukin-6 and Interleukin-10

C-reactive protein did not show a statistically significant change within the whole population. A statistically significant increase was noted in the CKD stage 3b group. Interleukins-6 and -10 remained largely unchanged, with a reducing trend noted in interleukin-10 in CKD stage 4, and an increasing trend in interleukin-6 in CKD stage 3b. No statistical significance was seen in terms of interleukin-6 and -10. Table 2 summarizes these results.

### 2.4. Endothelial Function and Vascular Response: E-selectin, P-selectin and Pulse Wave Velocity

A non-statistically significant increasing trend throughout the duration of the study was noted for both E-selectin and P-selectin. This trend was irrespective of CKD stage. Figure 2 and Figure 3 summarize these results. The pulse wave velocity and augmentation index remained unchanged, with a decreasing trend in the pulse wave velocity noted in patients with CKD stage 3b. Appendix A indicate both data and statistical significance relevant to both the endothelial function and vascular response.

## 3. Discussion

This pooled data analysis of the IRON-CKD and Iron and Heart RCTs allowed for further identification of trends in measures of oxidative stress and its sequela and a comparison between potential effects stratified by CKD stage.

Intravenous iron therapy has been associated in short-term studies with cellular oxidative stress. A rise in TBARS has previously been noted within 30 min following the administration of older intravenous compounds in non-dialysis-dependent CKD individuals, however, a longer-term effect was not seen [12]. The current pooled analysis reflects the results seen in the Iron and Heart and IRON-CKD trials, whereby no long-lasting oxidative stress effect was recorded. Indeed, modern intravenous iron compounds such as ferric derisomaltose represent more stable structures, thereby limiting the release of intravenous iron rapidly from their carbohydrate moiety and their propensity for the production of labile iron and hence subsequent oxidative stress generation [16]. The absence of a significant difference between the responses to treatments depending on CKD stage suggests that any pro-oxidation effect of intravenous iron is not CKD stage-dependent.

Evidence surrounding the inflammatory and immunological responses secondary to intravenous iron is contradictory; previous studies of patients on hemodialysis have reported inconsistent results. Martin-Malo identified an increasingly deleterious effect on mononuclear cells in patients on hemodialysis following the administration of intravenous iron, whereas Malindretos and colleagues reported no increased inflammatory markers following the administration of either iron sucrose or low-molecular-weight iron dextran [17,18]. Similarly, upon a comparison of oral against intravenous iron, murine models of iron deficiency anemia demonstrated an increase in interleukin-6 following the administration of intravenous iron (ferric carboxymaltose) that was not present in the animals treated with oral supplementation [19]. Such a phenomenon, however, has not been translated in vivo: interleukin-6 and tumor necrosis factor-α were increased significantly 10 months following initiation of iron treatment in 61 patients on hemodialysis irrespective of therapy with either oral or intravenous iron (saccharated ferric oxide) [20]. In the current pooled analysis, markers of inflammation remained largely unchanged, reflecting the trend seen in oxidative stress of the whole population. This was not the case forC-reactive protein in participants with CKD stage 3b. Interleukin-6, which represents a pro-inflammatory substance, increased slightly following intravenous iron in the CKD stage 3b group, whereas the anti-inflammatory interleukin-10 decreased in CKD stage 4. Interestingly, interleukin-6 is also involved in regenerative and anti-inflammatory actions [21]. As such, and given the limited numbers, an extrapolation into the direct effects of intravenous iron in the inflammation cascade cannot be made. The possible differential effect on CRP is a novel finding that warrants further research; however, similar limitations apply, especially upon considering other causes that may explain such an isolated rise. The possibility of a distinct effect per stage of CKD (which is also characterized by greater oxidative stress and inflammation as eGFR decreases) requires further investigation and characterization.

The impact of intravenous iron administration on endothelial function and atherosclerosis has prompted further research. Previous murine models demonstrated that iron sucrose administration was associated with an increase in adhesion molecules in proportion to an increase in oxidative stress [22]. Moreover, measurements of carotid artery intima-media thickness as a marker of early atherosclerosis in 60 patients receiving hemodialysis identified a significantly positive relationship between intima thickness and cumulative iron dose (*p* = 0.015) [23]. Selectins are upregulated in atherosclerosis and inflammation; administration of ferric carboxymaltose in patients with inflammatory bowel disease (*n* = 15) demonstrated a decrease in P-selectin expression leading to reduced platelet aggregation [24]. Similarly, a progressive analysis of 13 female patients with iron-deficiency anemia administered with ferumoxytol or low-molecular-weight iron dextran showed a decrease in P-selectin levels [25]. The current pooled analysis displayed no particular trend in P-selectin or E-selectin levels, irrespective of CKD stage, suggesting that any resultant changes following administration are small and may not lead to significant clinical effects. An increase in P-selectin could be considered prothrombotic. Nonetheless, it is important to acknowledge CKD as a state of platelet dysregulation, as evidenced by deficient reactivity in platelets previously demonstrated in vitro and a systematic review and meta-analysis that found an increased bleeding time and decreased platelet aggregation in patients with CKD [26,27]. The lack of a significant trend in pulse wave velocity and augmentation index measurements, as surrogate markers of aortic stiffness, would suggest that there is no clinically relevant impact following the administration of iron in non-dialysis-dependent CKD or atherosclerosis secondary to an increase in selectins. 

The clinical markers, transferrin saturation and serum ferritin increased in their mean and median values above the threshold, as previously set by the UK Renal Association and suggestive of adequate replenishment in both circulating and storage iron [28]. This suggests that a high dose of intravenous iron is efficacious in replenishing iron stores, even with a single administration (as per Appendix A). Hemoglobin concentration was not greatly affected. This may be due to the absence of anemia in a large percentage of the patients involved in these two trials, hence iron replenishment would not necessarily lead to a further increase in hemoglobin. Importantly, hemoglobin concentrations remained stable, potentially highlighting an anemia-preventative effect. Nonetheless, the presence of absolute iron deficiency in some participants despite correction by the end of the trial indicates the need for personalized administration of iron dependent on weight and hemoglobin as noted in the NIMO-CKD [29].

The results observed are suggestive of the mechanistic safety of high-dose intravenous iron administration using modern intravenous iron compounds in patients with non-dialysis-dependent CKD. This is important, especially when one considers that iron deficiency (alongside iron overload) can be deleterious to the mitochondrial machinery [7]. The results appear to compliment murine models previously discussed by Bhandari and colleagues, whereby iron therapy caused a reduction in oxidative stress vulnerability and may have provided improvement in mitochondrial function leading to improved cardiac and skeletal functional capacity [13]. Such results may aid in mechanistically explaining the improved cardiovascular outcomes observed in large-scale RCTs within CKD including PIVOTAL and FERWON-Nephro including patients from various stages of CKD [30,31]. 

This data pooled analysis has allowed us to maximize the data to be utilized, and thus permit further analysis by stratification and subgroup analysis. Nonetheless, this analysis has a number of limitations that are inherent to the original RCTs, and those include small numbers of participants, the presence of potential confounders (such as smoking and the use of over-the-counter antioxidant medications) and the absence of statistical power, especially within the subgroups. Moreover, the results related to oxidative stress may be limited through the use of a single biomarker, and therefore do not represent accurately the oxidant status as a balance between pro- and antioxidant substances. It is important to note that other biomarkers could be useful such as other markers of lipid peroxidation (e.g., F2-isoprostanes), advanced glycation products and oxidative DNA damage measures (e.g., 8-Oxo-2’-deoxyguanosine). These were not originally investigated, and hence no results were available for this pooled analysis. Moreover, no markers relevant to antioxidant activity were recorded. The timeframe of the included data does not address the potential of an ultra-acute change, occurring over minutes to hours; however, it provides reassurance relevant to any medium-term effect that could manifest clinically through adverse vascular and cardiovascular effects. Furthermore, upon considering mechanistic safety, it is important to note hypophosphatemia as a side effect of iron administration [32]. However, the two studies included did not collect any data relevant to phosphate concentration.

The results discussed provide further evidence surrounding the utility, efficacy and safety of modern intravenous iron preparations in the management of iron deficiency in non-dialysis-dependent CKD. Nonetheless, they are not conclusive. Chronic kidney disease represents a complex interplay between pro-oxidant and antioxidant reactions that could represent adaptations to increasing uremia and other complications. The progression of CKD and the presence of anemia or iron deficiency are associated with increased cardiovascular morbidity and mortality. This may be related to the worsening oxidative stress with advancing CKD. The earlier administration of intravenous iron in the presence of iron deficiency but not anemia may represent a protective barrier to worsening antioxidant status. Different pathophysiological states may have a differential reaction to intravenous iron. This has been recently explored in patients with anemia admitted to the intensive care unit when compared to healthy volunteers. Intravenous iron was associated with no increase in oxidative stress in patients admitted to intensive care, whereas an increase in oxidative stress was noted in healthy volunteers [33]. Intravenous iron (ferric carboxymaltose) on the other hand, in patients with severe COPD (*n* = 44), displayed a decrease in pro-oxidant agents with an increase in antioxidant capacity [34]. Such results indicate a differential effect on oxidative stress that is dependent on any prior disequilibrium between pro- and antioxidant function. This may explain the subtle differences demonstrated between the different CKD stages. Further studies are required in order to delineate the optimal management of iron deficiency in patients with non-dialysis-dependent CKD, especially its potential benefit to both mitochondrial biogenesis and oxidative stress. Moreover, further investigation of the impact of CKD stage on response to intravenous iron is warranted.

## 4. Materials and Methods

This is a pooled data analysis of two RCTsof high-dose intravenous iron (ferric derisomaltose (Monofer^®^), Pharmacosmos A/S; 1000 mg) in patients with non-dialysis-dependent CKD. IRON-CKD was an open-label explorative single-center study including 40 patients with non-dialysis-dependent CKD and iron deficiency examining the impact of different intravenous iron preparations and doses on markers of oxidative stress, inflammatory and endothelial response [15]. Ten patients received high-dose intravenous iron (1000 mg) and hence are included in the analysis. The Iron and Heart Trial was a multicenter double-blinded randomized placebo-controlled trial comparing placebo (intravenous 0.9% NaCl) to high-dose intravenous iron [14]. Twenty-six patients received 1000 mg of ferric derisomaltose and are incorporated into the analysis cohort. The inclusion/exclusion criteria of each study have been previously published [35,36]. The two RCTs employed a similar methodology and explored similar outcomes, with assessments at similar data points. The data from baseline, 1-month post administration and 3-months post-administration have been utilized for data pooling and analyses. 

Oxidative stress was measured through the measurement of TBARS, which represents a by-product of lipid peroxidation. Thiobarbituric acid reactive substances have been widely used in research and represent the end product of lipid peroxidation of polyunsaturated fatty acids leading to the presence of malondialdehyde secondary to the action of reactive oxygen species. An isocratic high-performance liquid chromatography technique was employed for the quantification of TBARS.

Inflammation was assessed through measurements of interleukins-6 (pro-inflammatory), interleukin-10 (anti-inflammatory) and C-reactive protein. Enzyme-linked immunosorbent assays (ELISA) were utilized for interleukin measurements (Thermo Fisher Scientific, Carlsbad, CA, USA). C-reactive protein was quantified using the Beckman Coulter Technology (Danaher Corp, Brea, CA, USA).

The vascular and endothelial functions were measured clinically through pulse wave velocity measurements and biologically via measurement and analysis of selectins. Selectins are cell adhesion molecules that facilitate the adhesion of leucocytes to platelets and endothelial cells. They are linked to inflammation and atherosclerosis [37]. The Enverdis Vascular Explorer (Enverdis GmBH Medical Solutions, Jena, Germany) was used to measure pulse wave velocity, and ELISA technology was utilized in the quantification of E-selectin and P-selectin. The human CD62P ELISA kit (P-selectin; Abcam, Cambridge, UK) and the mouse E-selectin/CD62E Quantikine ELISA kit (R&D Systems, Minneapolis, MN, USA) were used. 

Clinical markers including hemoglobin, transferrin saturation, serum ferritin and creatinine were analyzed as per local hospital pathology laboratory practice. The estimated glomerular filtration rate (eGFR) was calculated using the CKD-EPI 2009 calculation.

Statistical analysis was performed using Tableau 2022 (1.1) (Salesforce Company, Seattle, WA, USA). Descriptive statistics (mean (SD) and median (IQR)) have been used to identify trends. Analysis of variance (ANOVA) was used to determine statistical significance with a *p*-value of less than 0.05 representing significance. Stratification of participants was by CKD stage based on the eGFR. CKD stage 3a was defined as an eGFR between 45–59 mL/min/1.73 m^2^, CKD stage 3b as 30–44 mL/min/1.73 m^2^ and CKD stage 4 as eGFR 15–29mL/min/1.73 m^2^.

## 5. Conclusions

The results of this pooled analysis indicate that high-dose intravenous iron is not associated with a significant increase in the measures of oxidative stress or the measures of endothelial and inflammatory response biomarkers. Subtle but not statistically significant changes were noted based on CKD stage in terms of IL-6 and IL-10. A statistically significant trend was noted in terms of CRP in participants with CKD stage 3b, possibly suggesting a differential response. Consideration should be paid to the limitations of the pooled analysis that include small numbers and single biomarker characterization of oxidative stress. The signals identified are promising but not conclusive. Further studies are necessary to characterize the oxidant effect that intravenous iron and resultant iron deficiency resolution may cause and to identify whether the early resolution of iron deficiency utilizing intravenous iron would be beneficial. In addition, further research on the impact of intravenous iron dependent on the CKD stage would be key in tailoring treatment accordingly.

## Figures and Tables

**Figure 1 ijms-23-16016-f001:**
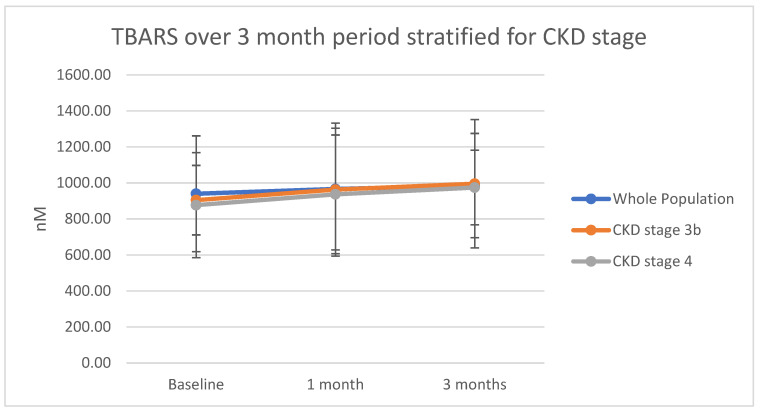
Thiobarbituric acid reactive substances in nM are plotted for the whole population and each individual large cohort of CKD (i.e., stages 3b and 4). Plotted points represent means, and error bars represent standard deviations.

**Figure 2 ijms-23-16016-f002:**
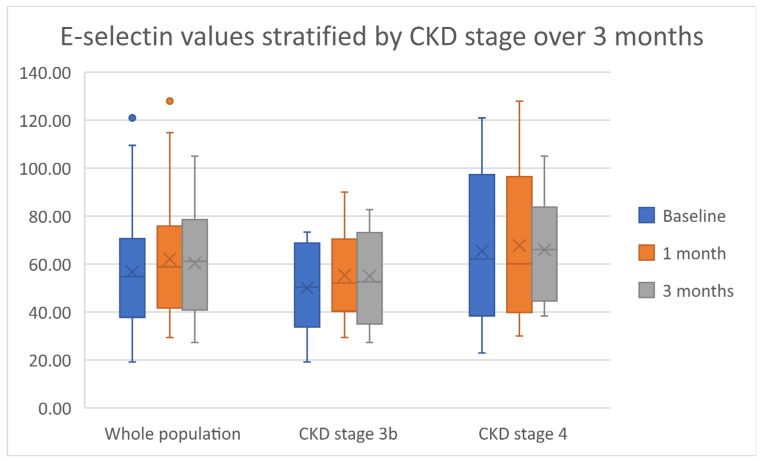
A boxplot representation of the data relevant to E-selectin. Dots represent outliers in the data.

**Figure 3 ijms-23-16016-f003:**
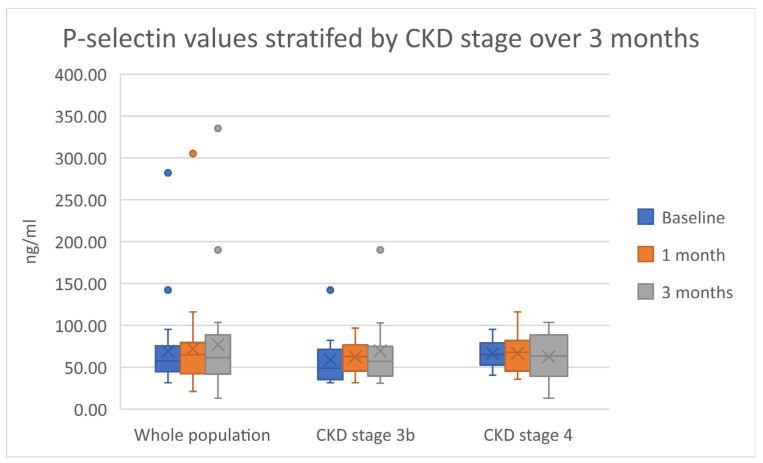
A boxplot representation of the data relevant to P-selectin. Dots represent outliers in the data.

**Table 1 ijms-23-16016-t001:** Baseline data.

Continuous Data		
Variable	Mean (SD)	Median (IQR)
Age/years	60.94 (12.27)	62.00 (15.00)
Hemoglobin/g/L	127.67 (11.24)	130.50 (10.25)
Serum Ferritin/μg/L	65.58 (38.57)	55.00 (36.50)
Transferrin saturation/%	21.19 (7.90)	19.00 (9.25)
Creatinine/μmol/L	185.78 (57.43)	172.00 (71.75)
Estimated glomerular filtration rate/mL/min/1.73 m^2^	32.77 (10.87)	32.63 (18.58)
Thiobarbituric acid reactive substances/nM	939.64 (321.49)	958.21 (304.45)
Interleukin-6/pg/mL	18.50 (31.23)	8.03 (2.04)
Interleukin-10/pg/mL	52.96 (75.24)	15.26 (31.48)
C-reactive protein/mg/L	6.52 (6.37)	4.00 (6.80)
E-selectin/ng/mL	56.74 (24.27)	54.86 (29.76)
P-selectin/ng/mL	69.26 (49.67)	57.50 (29.19)
Pulse wave velocity/m/sec	7.90 (2.98)	7.55 (2.15)
Augmentation index/%	23.97 (12.10)	25.00 (15.00)
**Categorical data**		
**Variable**	**Number**	**Percentage**
Gender		
Male	18	50.0
Female	17	47.2
Unknown (not speciified)	1	2.8
CKD Stage		
3a	3	8.3
3b	16	44.4
4	16	44.4
Unknown	1	2.8
Absolute iron deficiency		
Yes	18	50.0
No	18	50.0
Anemia		
Yes	10	27.8
No	25	69.8
Unknown	1	2.8
Primary cause of kidney disease		
Hypertension	8	22.2
Diabetes mellitus	8	22.2
IgA nephropathy	6	16.7
Congenital anomaly	1	2.8
Glomerulonephritis	1	2.8
HIV associated glomerulonephritis	1	2.8
Polycystic kidney disease	3	8.3
Chronic pyelonephritis	2	5.6
Tubular interstitial nephritis	1	2.8
Other/non-clear	5	13.9

**Table 2 ijms-23-16016-t002:** Markers of inflammation.

Whole Population (*n* = 36)				
	Baseline	1 Month	3 Months	*p*-Value
C-reactive protein/mg/L	6.52 (6.37)	5.85 (7.05)	9.21 (13.19)	0.27
Interleukin-6/pg/mL	18.50 (31.23)	22.02 (34.85)	22.35 (36.86)	0.74
Interleukin-10/pg/mL	52.96 (75.24)	52.79 (74.43)	51.04 (75.58)	0.93
**CKD stage 3b (*n* = 16)**				
C-reactive protein/mg/L	4.28 (4.24)	3.25 (1.84)	9.78 (8.97)	0.03
Interleukin-6/pg/mL	29.88 (45.38)	39.24 (51.86)	37.48 (51.38)	0.76
Interleukin-10/pg/mL	49.17 (70.66)	54.38 (82.61)	50.51 (76.59)	0.97
**CKD stage 4 (*n* = 16)**				
C-reactive protein/mg/L	8.54 (7.96)	8.14 (9.51)	9.67 (16.7)	0.79
Intereukin-6/pg/mL	9.31 (6.10)	11.09 (9.82)	9.29 (5.77)	0.95
Interleukin-10/pg/mL	54.46 (93.98)	46.55 (73.87)	43.59 (79.43)	0.80

Data presented as means (SD); statistical significance as per ANOVA testing. Not all participants supplied a blood test at every visit (there was no imputation for missing values).

## Data Availability

The data associated with this manuscript are not publicly available. They are available from the corresponding authors on reasonable request following permission and agreement by the sponsor of the studies: Research and Development Department—Hull University Teaching Hospitals NHS Trust.

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
