# Peer review of "Markers of Oxidative Stress, Inflammation and Endothelial Function following High-Dose Intravenous Iron in Patients with Non-Dialysis-Dependent Chronic Kidney Disease—A Pooled Analysis"

_ijms, 2022, doi:10.3390/ijms232416016_

Round 1

Reviewer 1 Report

I believe this paper cannot be considered for publication. It lacks several major points. 

Major issues

  1. Introduction. This section should be carefully reassessed. The objective of the study is not explained. Moreover, it seems quite too long and inconclusive. 

  2. Results. What is thiobarbituric acid? It is cited in the abstract but a full explanation on what this molecule is, how to measure it and why it is measured is lacking. 

  3. Results. Table 2. There are various reasons for C-reactive protein elevation in CKD patients. 

  4. Results. Again, what are E- and P-selectin? They are not described. 

  5. Discussion. Results do not support many of the discussion points raised. How can you draw conclusions from a paper with such a little study population?

  6. Materials and methods. This should be reorganized per points.

Author Response

Please see attachment for reply to all reviewers.

Thank you.

Reviewer 2 Report

In the current manuscript, the authors studied the effect of intravenous iron therapy on oxidative stress, inflammation, and endothelial function in CKD patients. The authors found that intravenous iron administration is safe and not associated with mentioned problems.

Comments to the authors

1)      Authors mentioned that intravenous iron administration is associated with hypophosphatemia. Several other studies also showed this adverse effect. Did the authors measure serum phosphate levels in patients after the iron treatment? That would be interesting to see in this manuscript.

2)      There is a discrepancy in the number of anemic patients in the text and table. Line 112 says that 9 (25%) were anemic. But table 1 shows that 10 patients are anemic. Moreover, the authors mentioned a total of 36 patients, and here, 10 anemic patients, 15 non-anemic patients, and 1 unknown, then what about the other 10 patients?

Author Response

(The authors gave the same response as above.)

Reviewer 3 Report

The authors mention the limitations of the study such as the sample number, the biomarkers analyzed, and the iron supplement administered. Therefore, I consider that the findings of this study do not answer the hypothesis or question that the authors want to resolve and, although the results are valuable, they should be expanded considering pro-oxidant enzymes, this system antioxidant, and other clinical markers such as lipoprotein oxidation and glycation products advanced.

Author Response

(The authors gave the same response as above.)

Round 2

Reviewer 1 Report

This paper could have been better designed and conducted. Limitations, which I believe are the most important thing of this paper, are now clearly enlisted for the benefit of readers. 

Reviewer 2 Report

The authors clarified the comments, and overall quality was improved

Reviewer 3 Report

The authors are encouraged to consider more oxidative stress and proinflammatory biomarkers in their future research to further discussion of the potential deleterious effects of iron supplementation.